# PathEX: Make good choice for whole slide image extraction

**Xinda Yang**[1☯]*, **Ranze Zhang**[2,3☯], **Yuan Yang**[4☯], **Yu Zhang**[1], **Kai Chen**[2,3,5]*

**1** Renmin University of China School of Information, Beijing, P.R. China, **2** Breast Tumor Center, Guangdong Provincial Key Laboratory of Malignant Tumor Epigenetics and Gene Regulation, Sun Yat-sen Memorial Hospital, Sun Yat-sen University, Guangzhou, Guangdong, China, **3** Breast Tumor Center, Sun Yat-sen Breast Tumor Hospital, Sun Yat-sen Memorial Hospital, Sun Yat-sen University, Guangzhou, Guangdong, China, **4** Department of Research and Development, Health Data (Beijing) Technology Co., Ltd, Guangzhou, Guangdong, P.R. China, **5** Artificial Intelligence Lab, Sun Yat-sen Memorial Hospital, Sun Yat-sen University, Guangzhou, Guangdong, China

☯ These authors contributed equally to this work.
* jasnei@163.com (XY); chenkai23@mail.sysu.edu.cn (KC)

## Abstract

### Background

The tile-based approach has been widely used for slide-level predictions in whole slide image (WSI) analysis. However, the irregular shapes and variable dimensions of tumor regions pose challenges for the process. To address this issue, we proposed PathEX, a framework that integrates intersection over tile (IoT) and background over tile (BoT) algorithms to extract tile images around boundaries of annotated regions while excluding the blank tile images within these regions.

### Methods

We developed PathEX, which incorporated IoT and BoT into tile extraction, for training a classification model in CAM (239 WSIs) and PAIP (40 WSIs) datasets. By adjusting the IoT and BoT parameters, we generated eight training sets and corresponding models for each dataset. The performance of PathEX was assessed on the testing set comprising 13,076 tile images from 48 WSIs of CAM dataset and 6,391 tile images from 10 WSIs of PAIP dataset.

### Results

PathEX could extract tile images around boundaries of annotated region differently by adjusting the IoT parameter, while exclusion of blank tile images within annotated regions achieved by setting the BoT parameter. As adjusting IoT from 0.1 to 1.0, and 1—BoT from 0.0 to 0.5, we got 8 train sets. Experimentation revealed that set C demonstrates potential as the most optimal candidate. Nevertheless, a combination of IoT values ranging from 0.2 to 0.5 and 1-BoT values ranging from 0.2 to 0.5 also yielded favorable outcomes.

**Data Availability Statement:** The code and associated documentation are uploaded to the GitHub repository. The repository URL is https://github.com/jasnei/PathEX.

**Funding:** the Natural Science Foundation of China (#82271650), Guangdong Science and Technology

Department (2020B1212060018) and Guangzhou
Science Technology and Innovation Commission
(#202102010221, #20212200003).

## Conclusions

In this study, we proposed PathEX, a framework that integrates IoT and BoT algorithms for
tile image extraction at the boundaries of annotated regions while excluding blank tiles within
these regions. Researchers can conveniently set the thresholds for IoT and BoT to facilitate
tile image extraction in their own studies. The insights gained from this research provide
valuable guidance for tile image extraction in digital pathology applications.

## Introduction

Whole slide image (WSI), also known as digital pathology, involves scanning pathological tis-
sue sections and subsequently converting them into high-fidelity digital images [1]. The tech-
nology provides medical practitioners an extensive and high-resolution approach to
anatomical assessment. The resultant digital assets are amenable to storage, transmission, and
computer-aided analysis, thereby enabling remote diagnostics, distance learning, and digital
curation [2]. It has become indispensable in modern clinical practice and is increasingly
regarded as a technological prerequisite within laboratory settings [3]. WSIs are characterized
by their exceptionally large image size and high resolution, often reaching dimensions of up to
100,000×100,000 pixels at a magnification of 40X. As the adoption of whole slide imaging con-
tinues to expand, there is a growing demand for effective and efficient gigapixel image
analysis.

Deep learning (DL) is at the forefront of computer vision, showcasing significant improve-
ments over previous methodologies on visual understanding [4]. This technology has found
application across various domains, including tissue segmentation, mutation prediction and
slide classification in WSI [5–9]. Artificial neural networks (ANNs) and convolutional neural
networks (CNNs) are capable of learning in both supervised and unsupervised manners.
Supervised learning relies on labeled data, such as histopathological diagnoses and manual
annotations, to instruct the network on detecting and classifying features in unknown datasets.
But the size and variability of manual annotations present challenges in training deep neural
networks.

WSI generally can be as large as 100,000×100,000 pixels at a 40X magnification, presenting
challenges in training deep neural networks end-to-end. Consequently, contemporary meth-
ods often adopt a tile-based approach, where WSIs are partitioned into numerous small image
tiles, from which models extract and aggregate features to make slide-level predictions [10–
13]. Tile images are usually square regions with dimensions ranging from $32 \times 32$ to
$10,000 \times 10,000$ pixels, with a majority of approaches employing tiles of around 256×256 pixels
[4, 14–16].

Grid tiling, which WSI is partitioned into multiple non-overlapping tile images in a grid-
like pattern from top left corner of the slide image, is the most common tiling strategy and it's
well understood. Tools such as Histolab [17], PyHIST [18], and SliDL [19] are capable of per-
forming holistic WSI grid tiling strategy.

However, the utilization of these tools presents challenges in the context of extracting anno-
tated tile images, particularly in the case of tumor regions. The irregular shapes and variable
dimensions of tumor regions complicate the process. Researchers have attempted to address
these difficulties by employing methods such as the eight-points method for extracting candi-
date tile images within the boundaries of annotated regions [20]. However, the limitations of
this method become apparent when dealing with concave annotated regions, where it fails to

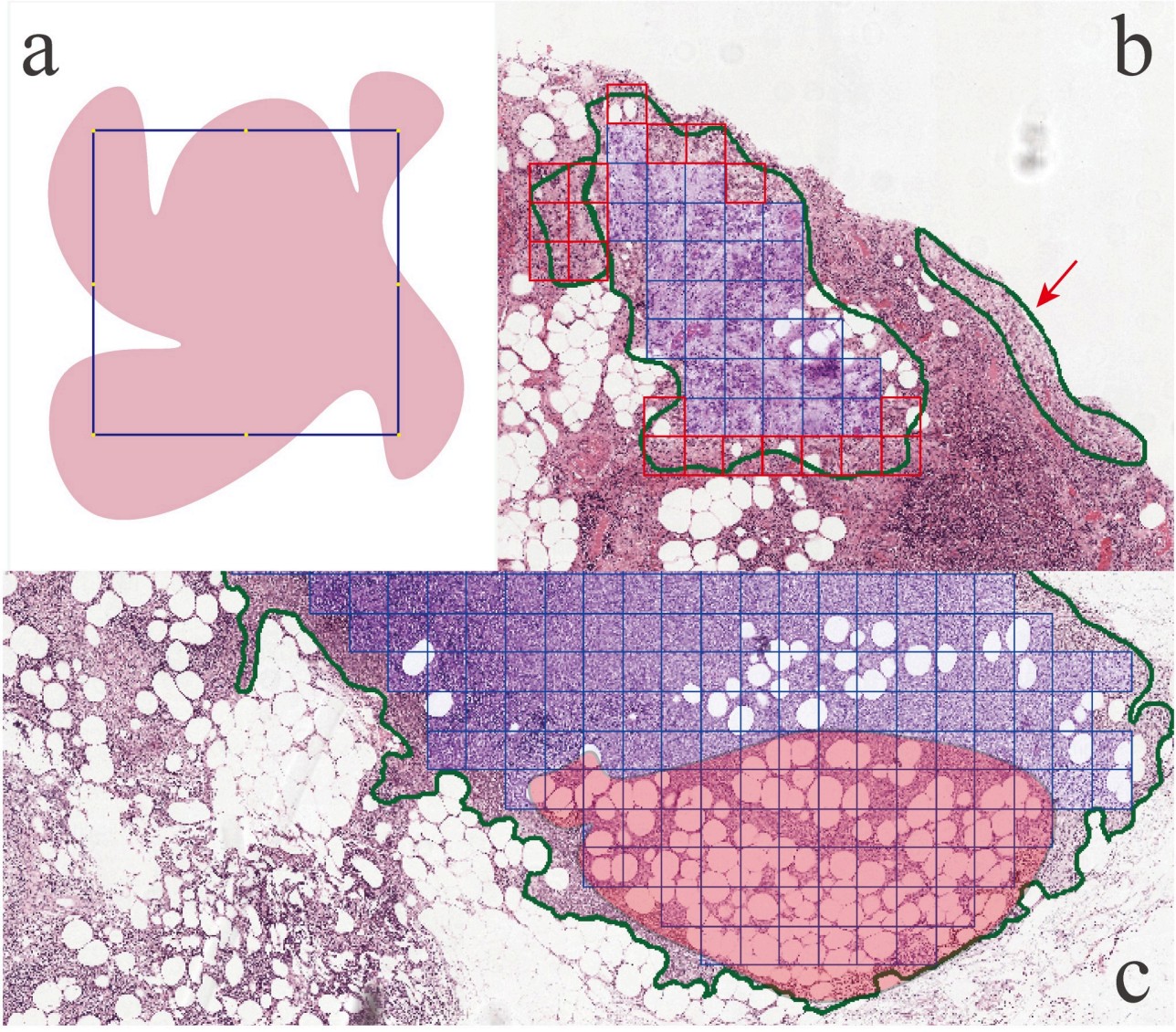

**Fig 1. Three types tile images we might overlook.** (a) Pink shaped region is annotated and the blue square is tile image, eight yellow dots shows eight-points, which shown eight-points method will not extract this tile image. (b) Green contour indicating the annotated region, and the blue squares are the candidate tile image. But the red squares and red arrow indicating that the tiles images are not candidate for extraction. (c) Green contour also indicating the annotated region, blue squire indicating candidate tile images. But some candidate tile image are lots of blank area or fat shown in the shaded region.

ensure the full containment of candidate tile images, shown in Fig 1a. This situation could be alleviated if extract tile images in a smaller tile size, but sometimes, smaller tile size would not be a good option.

Despite the fact that eight-points method can extract tile images inside the boundaries of the annotated regions mostly, however how to attain tile images which in the case shown in Fig 1a and how to attain tiles around the boundaries shown in Fig 1b still remain unsettled. Moreover, within the annotated regions, there are still some blank background area or area with lots of fat or others shown in Fig 1c. If we don't treat three types of noise tile images care-fully, we may result in inclusion of some irrelevant background information, noise tissue into DL training dataset, affecting the performance of neural network [21].

In order to solve the above problems, we first introduced the concept of intersection over tile (IoT) for handling the extraction problem of tile images around boundaries of annotated regions, particularly in cases where the region was concave. Additionally, we proposed the concept of blank over tile (BoT) to mitigate issues related to tile images containing substantial blank areas. Subsequently, we incorporated our algorithm for tile image extraction. In the dend, we extracted tile images into datasets for evaluating the performance of DL model, and determined the most effective combination of *IoT* and *BoT*.

## Related work

Several tools have been developed which provide a variety of functions. While providing a complete description and comparison of these tools is beyond the scope of this manuscript, here we briefly describe these tools and whether they can be used to extract tile images using manual annotation files or overall WSI extraction.

Histolab [17] is a Python library, offers a range of functionalities aimed at facilitating tissue analysis. These functions include classical image analysis techniques for tissue detection, artifact removal, and slide tiling. The library also supports the extraction of tile images from WSIs and includes its own tiling strategies, such as grid tiling, score tiling, and random tiling.

PyHIST [18] is a semi-automatic python command-line tool for WSI segmentation and preprocessing, producing a mask for input WSI and generating tile image with mask. However, same as Histolab, the tiling method requires a mask function to localize the tile, which entails computationally expensive intersection calculations between the tile and the mask.

MONAI [22] is an extensive Python library which is part of the PyTorch ecosystem and is designed as a unified framework for performing deep learning on medical imaging data. MONAI supports tiling of WSIs and provides extensive support model evaluation metrics. However it still does not provide tiling of WSI using manual annotation.

SliDL [19] is a Python library with a straightforward API for artefact and background detection, tile extraction from annotations, as well as model training and inference on WSIs. SliDL enables the extraction of tiles from annotations, which is particularly valuable for learning applications. It will extract all tile images within the annotations even if the tile image is background.

In contrast to the aforementioned tools, the proposed approach abstracts ideas into algorithms that could be applied to a broader range of applications, tiling for holistic WSI and for manual annotation files. With the proposed approach, we could control inclusion of how much the tiles intersection with the boundaries of annotated regions, and exclude the tile image based on the blank area of tile image.

## Proposed method

### Intersection over tile image

We have analyzed the tile images which are not extracted around the boundaries of the annotated regions. It is because the tile images are not 100% within the annotated regions, and these tile images are partial within the region, partial out of the regions, which indicate tiles could be a noise tile images. Similar to the eight-points method, in their research, they considered these tile images as noise images and they should be excluded. However, these noise tile images still partial within annotated region, with partial positive tissue, so it might be useful for training our DL model. Intuitively, to tackle this problem, we could use the proportion of intersection area between the tile image and the annotated region to decide whether to extract the tile images around boundaries or not. Intersection over union (IoU) [23] is commonly used to measure the accuracy of object detection segmentation. Inspired by IoU, we proposed

**intersection over tile (IoT)**, which was intersection of tile image and annotated region over the area of tile image.

$$\text{IoT} = \frac{\text{area}(A \cap T)}{\text{area}(T)}$$

Where $A, T \subseteq \mathbb{S} \in \mathbb{R}^n$, $A$ represents arbitrary annotated region, and $T$ is for tile image. We first calculate the area of $T$. Latter we would compare $T$ and $A$, whether these two polygons were intersecting, if they intersected, then we will calculate the area of the intersection. Finally, we would calculate a proportion between the area of the intersection and the area of the tile. The calculation of $IoT$ is summarized in Algorithm 1

**Algorithm 1**: Intersection over Tile

**Input:** Two arbitrary polygons: $A, T \subseteq \mathbb{S} \in \mathbb{R}^n$, $A$ for annotation, $T$ for tile

**Output:** IoT

1: $S_T$ = area ($T$)
2: state ← intersects ($T$, $A$)
3: **if** state **then**
4:     $S_{\text{intersection}}$ = area ($T \cap A$)
5: **else**
6:     $S_{\text{intersection}}$ = 0
7: **end if**
8: IoT = $\frac{S_{\text{intersection}}}{S_T}$

$IoT$ as a metric has the following properties:

1. $IoT$ is invariant to the scale of tile image.

2. $IoT$ is non-negative, ranging from 0 to 1.0.

    (a) The value 1 occurs for $IoT$ only when the tile image intersects with the annotated region perfectly, this only happens when the tile image is within or touches the annotated region from inside. i.e. $T \cap A = T \Rightarrow T \subseteq A$

    (b) The value 0 occurs for $IoT$ only when the tile image does not intersect with the annotated region or touches the annotated region from outside. i.e. $T \cap A = \varnothing \Rightarrow T \nsubseteq A$.

    (c) The value is in (0, 1.0), when the tile image intersects with the annotated region. i.e. $T \cap A = S$ and $S \neq \varnothing$

## Background area over tile image

For some tile images, although they were within the annotated regions, having a substantial amount of blank spaces or fats in the tile images, could render them unsuitable for use as the positive tile images. These areas, which we termed as the background, may have very little tissue or no cell at all. To address this issue, we introduced **background over tile (BoT)**, similarly as $IoT$. BoT referred to the proportion of blank or background area over the entire tile image, which determined whether the tile image was eligible for extraction. Given a tile image, $T \in \mathbb{R}^n$, the tile image was an RGB image normally. To begin with we could apply Gaussian blur with a $5 \times 5$ kernel size or similar blur algorithm. Next, we would convert the blurred image from RGB color space to HSV color space and obtained a background mask using the HSV threshold setup. The background mask might have some minor defects, thus, we further refined it by applying a dilation operation with a $3 \times 3$ structuring element. On the refined background mask, we calculated the area of the mask which is the contour area of the mask.

Finally, we would get the *BoT* by dividing the area of the background over the area of the tile image which is product of height and width of tile image. The calculation of *BoT* was summarized in Algorithm 2

$$\text{BoT} = \frac{\text{area}(B)}{\text{area}(T)}$$

Where $B, T \subseteq \mathbb{S} \in \mathbb{R}^n$, $B$ is for background of a tile image, and $T$ is for tile image. We do not use BoT directly, we would use $1 - \text{BoT}$, which means **tissue over tile image (ToT)**.

*BoT* as a metric has the following properties:

1. *BoT* is invariant to the scale of tile image.

2. Similar to *IoT*, *BoT* is non-negative, ranging from 0 to 1.0.

   (a) The value 1 occurs for *BoT* only when the tile image is full with background, which means there is no any tissue.

   (b) The value 0 occurs for *BoT* only when the tile image is all covered with tissue.

   (c) The value is in range (0, 1), when there is some background and tissue on tile image, this would be most common case for all tile image.

**Algorithm 2**: Background over Tile

```
Input: A tile: T ⊆ S ∈ ℝⁿ
Output: BoT
1: S_T = area (T)
2: Gaussian blur, 5 × 5 kernel size on T
3: Convert RGB space to HSV space
4: Get background mask from HSV space
5: Dilate with 3 × 3 structuring element on mask
6: S_bg = area (background)
7: BoT = S_bg/S_T
```

## PathEX: WSI tiling algorithm

According to the properties of both *IoT* and *BoT*, a floating threshold for both metrics would be required to determine tile image extraction. We incorporated *IoT* and *BoT* with some other operations into an algorithm which would extract tile images from holistic WSI and manual annotation files (ANO). Annotated region extraction would require an annotation file (ANO), which contained coordinates of the annotated region, while holistic WSI extraction would not require an ANO.

**For the annotated region extraction.** Holistic WSI tile image extraction would be using *BoT* only, while annotated region extration required both *IoT* and *BoT*. We would specify the annotated regions extraction in the following. In the beginning, we parsed the ANO to get coordinates. Subsequently, we partitioned WSI into the intended tile size and got the locations of the tile image. In the following, we would iterate all the locations, and calculate *IoT* and *BoT* respectively, if both metrics met the thresholds respectively, then the tile image would be extracted. WSI tiling algorithm (**PathEX**) was summarized in Algorithm 3.

**Algorithm 3**: Tiles extraction

```
Input: ANO, WSI, IoT_thresh and BoT_thresh ∈ [0, 1], W for WSI, A for Anno-
       tated coordinates, A, W ⊆ S ∈ ℝⁿ
Output: Tiles
1: parse ANO get A
2: locations_tiles = partition (W)
```

```
3: for loc ← locations_tiles do
4:    IoT = calculate_iot(A, loc)
5:    if IoT ≥ IoT_thresh then
6:       T = get_tile(loc)
7:       BoT = calculate_bot(T)
8:       if 1 − BoT ≥ BoT_thresh then
9:          Save tile
10:      end if
11:   end if
12: end for
```

As aforementioned, we incorporated *IoT* and *BoT* into algorithm so that we could extract tile images around boundaries of annotated regions and exclude the background tile images within annotated regions. One idea, which would be excluding both tile images around the boundaries and background tile images, would be better for comprising a clean nice datasets for training DL model. In this work, however we found some interesting results in section.

## Experiments

### Implementation

We implemented the algorithm using the Python library, version 3.9. Openslide [24] and tiff-file [25] for low-level WSI operations, Numpy [26] for fast numerical computations, Shapely [27] for geometry processing, and OpenCV [28] and Pillow [29] for image processing algorithms.

### Dataset

**CAMELYON16.** CAMELYON16 (CAM) [30] was a challenging aim to automated detection of metastases in hematoxylin and eosin (H&E) stained WSI of lymph node sections. The data in this challenge contains a total of 400 WSIs of sentinel lymph node from two independent datasets collected in Radboud University Medical Center (Nijmegen, the Netherlands), and the University Medical Center Utrecht (Utrecht, the Netherlands). The training set comprised 111 slides containing metastases and 159 normal slides, while testing set includes 129 slides with metastases and normal. The slides are all 40x magnification with handcraft annotation around the positive region. Four our experiment, we used 128 out of 159 negative and 111 positive slides for the comprising training set. 12 negative and 36 positive slides out of 129 are for the testing set.

**PAIP.** The PAIP 2021 (PAIP) challenge involved multiple organ cancers, including colon, prostate, and pancreatobiliary tract, with 150 training slides, 30 validation slides, and 60 testing slides. Slides are at 20x magnification. The challenge aims to promote the development of a common algorithm for automatic detection of perineural invasion in resected specimens of multi-organ cancers. The manual annotation regions are mostly rectangle within the designated type of tissue region and are different from CAM dataset. We took colon slides for our experiment. There are a total of 50 slides with annotation of colon cancer, 40 slides are for training set, and 10 slides are for testing set. Each slide has four layers annotations (nerve without tumor, perineural invasion junction, tumor without nerve, non tumor without nerve), which is rectangle annotations. We extracted tumor without nerve as the positive, and non-tumor without nerve as the negative, both together were for a two-classification dataset.

**Training set.** We incorporated *IoT* and *BoT* into tile extraction, and the tiles extracted were for training a classification model. We extracted tile image in size $512 \times 512$ pixels for both CAM and PAIP datasets, instead of common $256 \times 256$ pixels. For the CAM negative slides, we randomly extracted 10% tile images of each slide. And, for positive slides, we set up

**Table 1. Set up of *IoT* and 1 -*BoT* for tile images extraction of both training and testing set, number of tile images for both CAM and PAIP dataset.**

| | Set | IoT | 1—BoT | CAM | | PAIP | |
|---|---|---|---|---|---|---|---|
| | | | | negative | positive | negative | positive |
| training | A | 0.1 | 0.0 | 114,668 | 78,890 | 16,779 | 22,734 |
| | B | 0.2 | 0.0 | 114,668 | 75,858 | 15,969 | 21,206 |
| | C | 0.2 | 0.2 | 114,668 | 75,022 | 10,841 | 21,101 |
| | D | 0.2 | 0.5 | 114,668 | 72,268 | 8,525 | 20,166 |
| | E | 0.5 | 0.2 | 114,668 | 68,663 | 9,511 | 16,943 |
| | F | 0.5 | 0.5 | 114,668 | 66,760 | 7,515 | 16,279 |
| | G | 1.0 | 0.2 | 114,668 | 57,292 | 6,940 | 10,943 |
| | H | 1.0 | 0.5 | 114,668 | 55,942 | 5,465 | 10,613 |
| testing | | 0.3 | 0.3 | 8,453 | 4,623 | 1,816 | 4,575 |

different *IoT* and *BoT* thresholds for tile images extraction for experiment. The threshold for *IoT* was varied at 0.1, 0.2, 0.5, and 1.0, corresponding to at least intersection percentages of 10%, 20%, 50%, and 100% with annotated regions would be inclusion for comprising positive data. Similarly, the 1—*BoT* threshold was set at 0.0, 0.2, and 0.5, denoting tile images inclusion with 0%, 20%, and 50% tissue content, respectively. Due to the WSI procedure, it was common that area without tissue or with fat is across the holistic WSI. We did not set 1—*BoT* to 1.0, as 1.0 indicating inclusion 100% tissue content, which would reduce the tile images quite a lot. In this point, we kept 1—*BoT* maximum at 0.5. All PAIP training set slides were also extracted tile images with these combinations. With these set up, we would have 8 combinations as following Table 1, resulting in a total of 8 training sets and 8 models for each dataset.

**Hold-out testing set.** In the actual case, we need to identify all tile images of holistic WSI. And in our experiment, we would not use an uncertain threshold for the testing slides, so we specified the *IoT* threshold of 0.3 and 1—*BoT* of 0.3. Both thresholds were slightly different from the training set to suitably evaluate the model's performance. After these settings, we extracted 4,623 tile images from 36 positive slides of the CAM testing set, and 8,453 tile images from 12 negative slides of the same set, employing identical tiling strategy as those used during training set. From 10 slides of the PAIP dataset, we got 1,816 negative and 4,575 positive tile images. These test sets were hold-out to evaluate the model performance. Detail summarized in Table 1.

## Training protocol

During the model training phase, the ResNet-50 [31] architecture was utilized as the backbone, and optimization was performed using the Adam optimizer [32] with a weight decay value of 0.00002 and betas set to (0.9, 0.99). The training procedure was carried out on a computing infrastructure consisting of 8 Geforce RTX 3090 GPUs, employing a batch size of 80 for each GPUs. The initial learning rate was set at 0.0001, and a linear warm-up cosine decay schedule with 5 epochs was implemented. Mixup [33] alpha at 0.15 was applied during batch processing and discontinued after 60 epochs, with cross entropy serving as the loss function [34]. The total number of training epochs was set to 100. The training data was partitioned such that 80% of each class was used for training, and 20% of each class was allocated for validation.

**Augmentation.** A stochastic data augmentation module that randomly transforms any given data example which is denoted *x*. Data augmentation module include normal augmentation and strong augmentation.

Normal augmentations include the following: (1) Randomly crop and resize back to the original size. (2) Randomly choose between vertical horizontal flips. (3) Randomly choose between Gaussian and ISO noise injection. (4) Randomly gamma transformation. Normal augmentation is applied to every training iteration.

Strong augmentations include the following: (1) Randomly choose one of color jitter, random brightness with contrast and image to gray. (2) Randomly choose one of Gaussian blur, defocus and motion blur. (3) Affine transform. (4) Salter and pepper noise injection. (5) Coarse dropout. (6) Random grid shuffle. (7) Elastic transform. (8) Sharpen. (9) Image compression. For all transforms mentioned above, we will randomly sample two strong augmentations during each training iteration to avoid too intensive augmentation.

All the augmentations were done by albumentations library [35]. We would train all datasets generated with Algorithm 3 (PathEX) following the aforementioned protocol.

## Result

### Time required for tiling

Firstly, we reported the processing time for extraction of tile images by algorithm PathEX. *IoT* was set to 0.1 and 1—*BoT* was set to 0.0 same as set A (mentioned in section.). Total time in processing 111 positive slides was 1,729 seconds, which was approximately equivalent to 28 minutes. The variability in processing time stemmed from the diverse dimensions of the manually annotated regions, with the minimum processing time per slide being 3.24 seconds, the maximum being 95.19 seconds, and the mean being 15.58 seconds per WSI. Total 78,890 tile images were extracted. All processing time was summarized in Table 2.

For PAIP 50 slides, tumor without nerve layer was extracted in the same configuration as CAM dataset. The total processing time was 339 seconds, which was less than 6 minutes, and the average time for each slide was 6.78 seconds. Processing time was less than CAM dataset, mainly because the first PAIP dataset is 20x and the second annotated region is smaller than that of CAM dataset. 22,734 positive tile images were extracted. Processing time detail was summarized in Table 2.

### Annotated region extraction

PathEX could extract tile images around boundaries of annotated region, if we set up the *IoT* properly. Shown in Fig 2 by adjusting *IoT*, we could extract tile images around the boundaries differently. Setting *IoT* from lower to higher, we could extract less tile images. For instance, if we wanted to extract tile images 100% within annotated areas, we could set *IoT* at 1.0, shown in Fig 2a. However, if we wished to extract all tile images mostly around the annotated boundaries, we could lower *IoT* setting to 0.1, shown in Fig 2b, in this way we could extract mostly the tile images. If we wanted to extract more tile images, but with less noise tile image, we could set *IoT* a little bit higher, such as 0.2, 0.5, etc, shown in Fig 2c and 2d.

Furthermore, the exclusion of blank tile images within annotated regions could be achieved by setting parameter for *BoT*, effectively eliminating irrelevant or noise-containing tile images, shown in Fig 3, With *BoT* setting at 0.1, meaning that tile image with 90% blank area would be

**Table 2. Processing Time for CAM 111 positive slides and PAIP 50 slides, unit is in seconds.** Q1 is 25th percentile, Median is 50th percentile, Q3 is 75th percentile.

|  | min | max | mean ± std | variance | Q1 | Median | Q3 | Total |
|---|---|---|---|---|---|---|---|---|
| CAM | 3.24 | 95.19 | 15.58 ± 14.16 | 200.61 | 10.47 | 11.41 | 13.6 | 1729.48 |
| PAIP | 2.46 | 15.99 | 6.78 ± 2.75 | 7.57 | 4.99 | 6.32 | 7.6 | 339.16 |

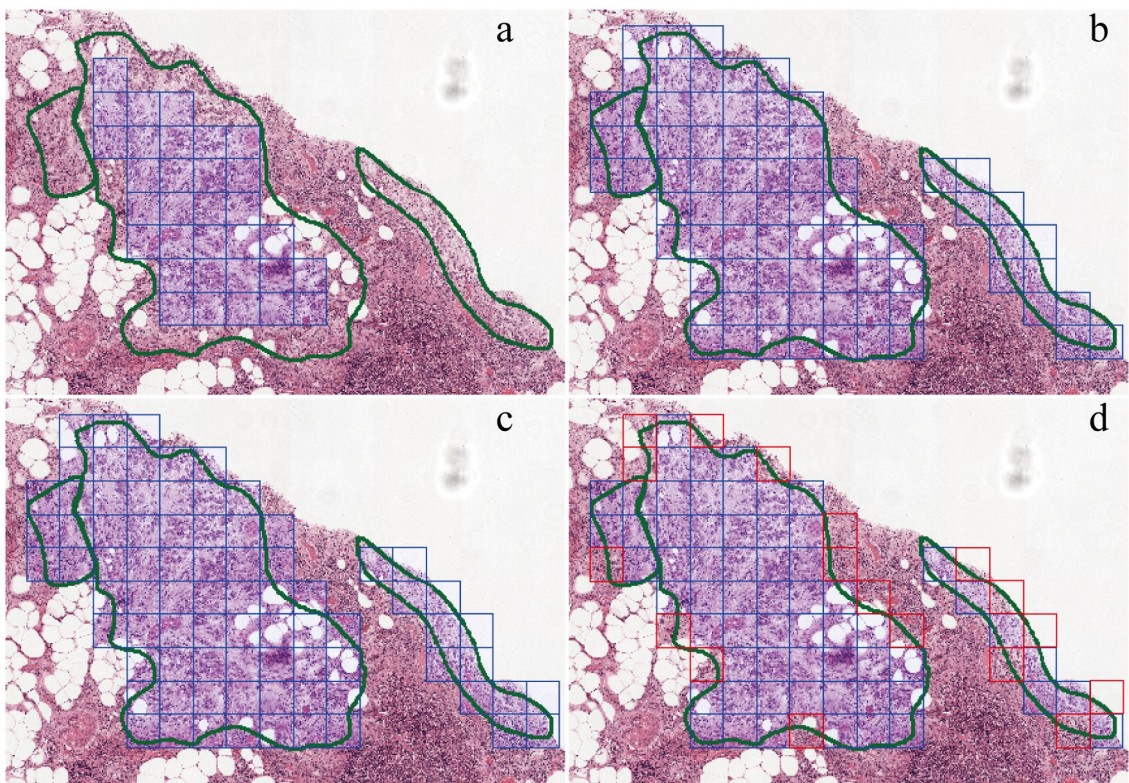

**Fig 2. Visualization for boundaries extraction.** Blur square are tile images we will extract. (a) *IoT* set at 1.0, square tile images are 100% within annotated regions, while there are some annotated region cannot be extracted. (b) *IoT* set at 0.1, greater than 10% intersecting area will be included. (c) *IoT* set at 0.2, which only very few difference between (b). (d) *IoT* set at 0.5, indicating inclusion tile image greater than 50% intersecting area. The red square is the tile not included compared wiht (b).

excluded (Fig 3b), showing few tile images not included. As we set up a little bit higher the *BoT*, more tile images would be excluded, shown in Fig 3c and 3d, *BoT* set at 0.2 and 0.5, respectively.

The combined utilization of *IoT* and *BoT* in PathEX provides a flexible approach to extract tile images around boundaries and exclude blank tile images, demonstrating its adaptability across diverse applications.

## Results for training model

Shown in Table 1. As *IoT* set up from 0.1 to 1.0, and 1—*BoT* from 0.0 to 0.5, we had 8 combinations of these two parameters. We got 8 training sets from A to H. To find out the best combination of *IoT* and *BoT*, we evaluated performance of model trained on each training set and hold-out test set. We mainly looked at the accuracy, but also with precision, recall and f-score (F1 score), specificity. For each train set, we would look at the best accuracy of validation set, and the trained model would run inference on hold-out test set to get evaluation metrics. We have some findings as below:

**Higher *IoT* and 1—*BoT*, less noise tile images.** The number of positive tile images we got was decreasing, shown in Fig 4a, as we set up *IoT* and 1—*BoT* from lower to higher, revealing a reduction in the number of noisy tile images with some positive tissue and other tissue. The number of positive tile images extracted from set A was found to be the highest notably,

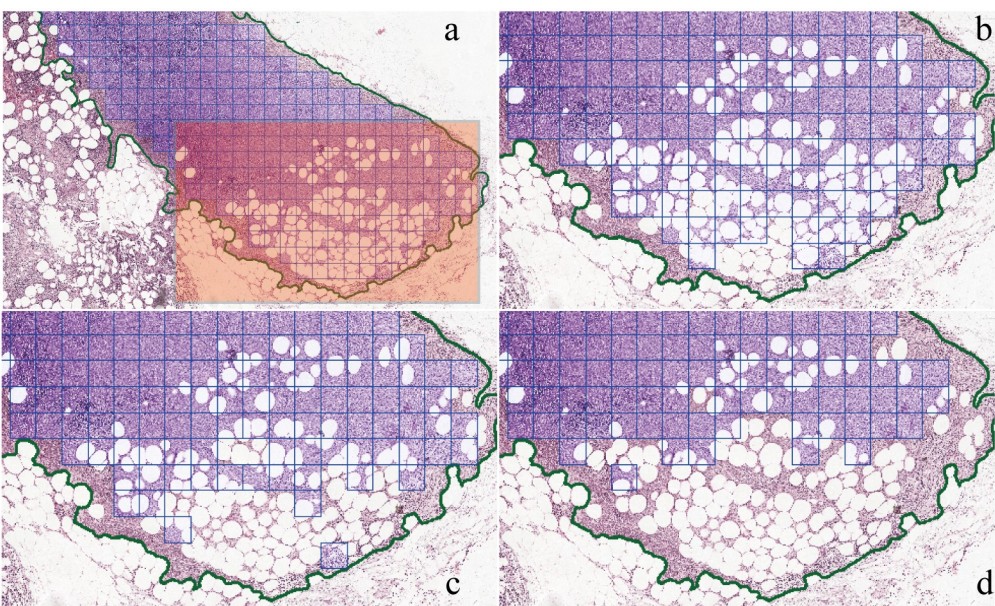

**Fig 3. Visualization exclusion blank tile image within annotated region.** Blue square tile images are extracted. (a) Showing in the red shaped area, blank tile images also included, b, c and d are zoom-in of the red shaped area. (b) *BoT* set at 0.1, showing that if the tile image is 90% blank area will excluded. (c) *BoT* set at 0.2, tile image with 80% blank area will be excluded, showing that more tile images exclude from a, b. (d) *BoT* set at 0.5, showing a lot more tile images at the red shaped area are not included.

while set H yielded the least. Furthermore, a comparison between set H and set A revealed a 30% and 47% reduction in tile images in CAM and PAIP, respectively. Additionally, the transition from set A to set H showed an overall improvement in all metrics alongside a decrease in noise tile images, indicating reduced uncertainty in the training and validation sets, shown in Fig 4b. The Pearson correlation coefficient Table 3 confirmed a strong negative correlation between the reduction in noise tile images and the metrics of the validation set on CAM dataset, with accuracy ($r = −0.8918$), precision ($r = −0.8503$), recall ($r = −0.7565$) and f-score ($r = −0.8476$), and all p-values less than 0.05. The Pearson correlation coefficient revealed a strong correlation in test set on PAIP dataseet, which showed in Table 4.

**Higher *IoT* and 1—*BoT*, not better performance in test set.** As mentioned above, higher *IoT* and 1—*BoT*, we could get less uncertainty dataset, which model trained might perform well in validation set. However model might not perform well on the hold-out test set, which obviously showed over fitting. We visualized the difference of accuracy between validation set and the hold-out test set (z-score) shown in Fig 4d for CAM dataset and Fig 5d for PAIP dataset. For instance, the greatest difference of accuracy was up to 7.05% at set G, whereas the least was 4.96% at set C where *IoT* was 0.2 and 1—*BoT* was 0.2 for CAM dataset. Model trained on set C showed the least over fitting on test set for CAM dataset, and achieved better results in test set, though was not the best, set F was a little better than set C and was the best for CAM dataset. And the model train on set C from PAIP dataset showed similar result. Difference of accuracy for set C was about 2.098% which was second least in all eight sets and the accuracy was also second best. Both metrics only a little difference from the best during all eight sets. From the results obtained with these two datasets, we could conclude that set C was probably the best candidate for tile images extraction. Although we experimentally found that set C may be the best candidate. But we believed that the combination of *IoT* from 0.2 to 0.5 and 1—*BoT* from 0.2 to 0.5 would also be a nice trial.

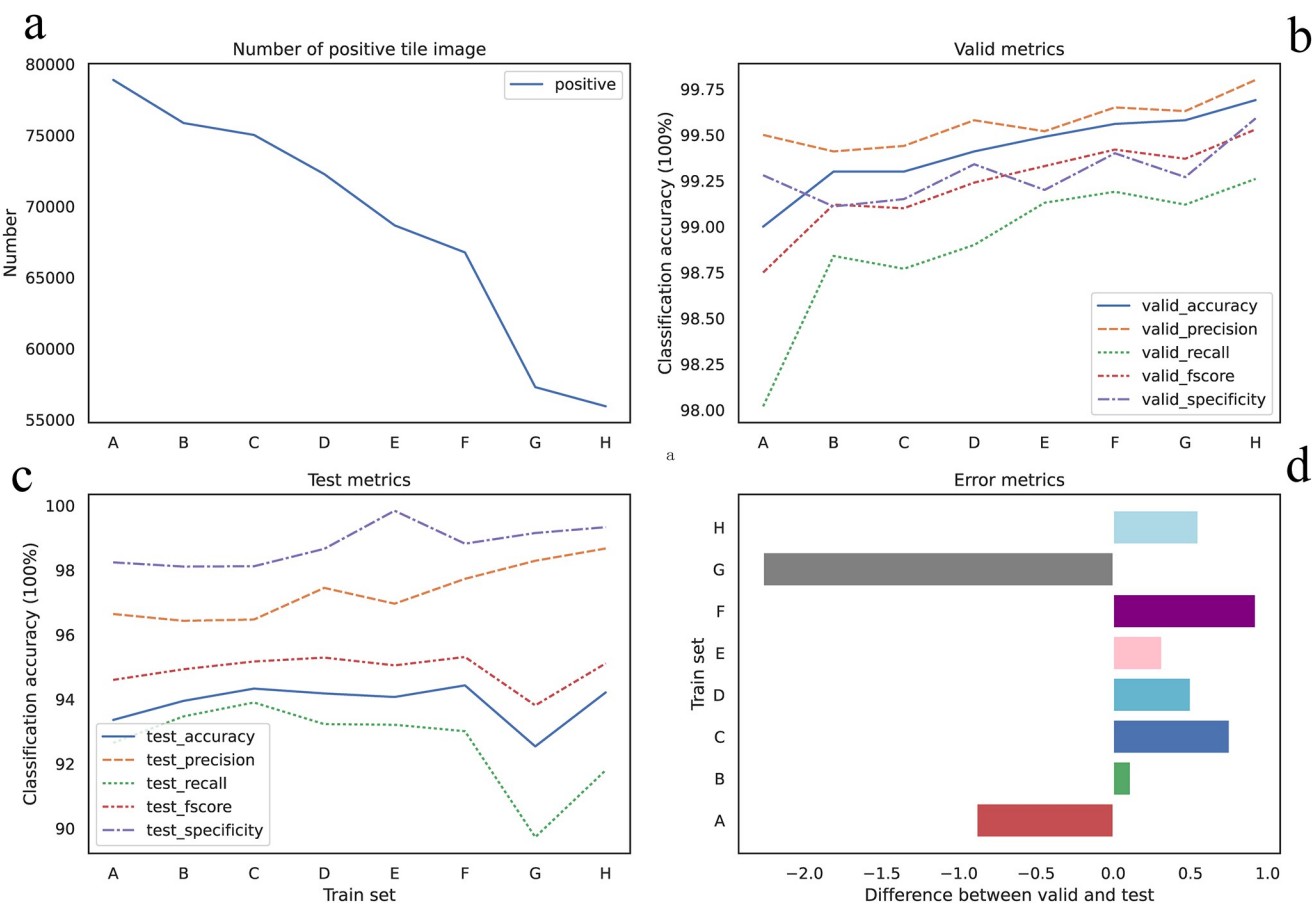

**Fig 4. Metrics for CAM dataset.** (a) Number of the positive tile images from set A to set H is decreasing. (b)Visualize performance model on validation set. (c) Visualize performance on hold-out testing set. (d) The relative difference in the performance between validation set and test set (z-score).

**Reducing 100% blank tile image, improve performance.** The difference between set B and set C, was that $1—BoT$ threshold was different. 1- $BoT$ for set B was 0 indicating that set B would have some 100% blank tile images. While, 1- $BoT$ for set C is 0.2, showing that there was not any 100% blank tile images in the set C. Shown in Figs 4d and 5d, taking accuracy as a proxy metric, there was some improvement from set B to set C in both datasets, which indicated reducing 100% blank tile images would benefit the model performance. We should probably try to void including 100% blank tile images into our positive set in our work, as such kind tile image was definitely noise.

**Some noise tile images in dataset, improved robust performance.** As we mentioned in section., noise tile image is a tile image which image is partial positive and partial negative. In both CAM and PAIP datasets, from set A to set H, less noise tile images would reduce the quantity of tile images in total, especially in PAIP datasets, both negative and positive tile image were decreasing. However the less tile images we got, most likely that the model we trained was easier over fitting the data. While, with some noise tile images the quantity of the dateset would be larger, for instance set C, the model showed less over fitting indicating more robust model trained on the set C. We could consider extract some noise tile images in our research, especially when the dataset was too small.

**Table 3. Pearson correlation coefficient for both validation set and testing set of CAM datasets.**

|                |             | number of positive |
|----------------|-------------|--------------------|
| **valid accuracy**  | coefficient | -0.8918**       |
|                | p-value     | 0.0029             |
| **valid precision** | coefficient | -0.8503**       |
|                | p-value     | 0.0075             |
| **valid recall**    | coefficient | -0.7565*        |
|                | p-value     | 0.0298             |
| **valid fscore**    | coefficient | -0.8476**       |
|                | p-value     | 0.0079             |
| **test accuracy**   | coefficient | 0.2422          |
|                | p-value     | 0.5633             |
| **test precision**  | coefficient | -0.9367**       |
|                | p-value     | 0.0006             |
| **test recall**     | coefficient | 0.7543*         |
|                | p-value     | 0.0305             |
| **test fscore**     | coefficient | 0.3108          |
|                | p-value     | 0.4536             |

*$p < 0.05$
**$p < 0.01$

## Discussion

In our study, we proposed the PathEX algorithm, a novel approach that combines *IoT* and *BoT* algorithms. The primary objective of PathEX is to efficiently extract tile images from the boundaries of annotated regions while excluding blank tile images within these regions. Our

**Table 4. Pearson correlation coefficient for both validation set and testing set of PAIP dataset.**

|                |             | number of positive |
|----------------|-------------|--------------------|
| **valid accuracy**  | coefficient | -0.5518         |
|                | p-value     | 0.1561             |
| **valid precision** | coefficient | -0.3026         |
|                | p-value     | 0.4664             |
| **valid recall**    | coefficient | -0.6364         |
|                | p-value     | 0.0898             |
| **valid fscore**    | coefficient | -0.6742         |
|                | p-value     | 0.0667             |
| **test accuracy**   | coefficient | 0.9433**        |
|                | p-value     | 0.0004             |
| **test precision**  | coefficient | -0.8509**       |
|                | p-value     | 0.0074             |
| **test recall**     | coefficient | 0.9714**        |
|                | p-value     | 0.5702e-4          |
| **test fscore**     | coefficient | 0.9474**        |
|                | p-value     | 0.0004             |

*$p < 0.05$
**$p < 0.01$

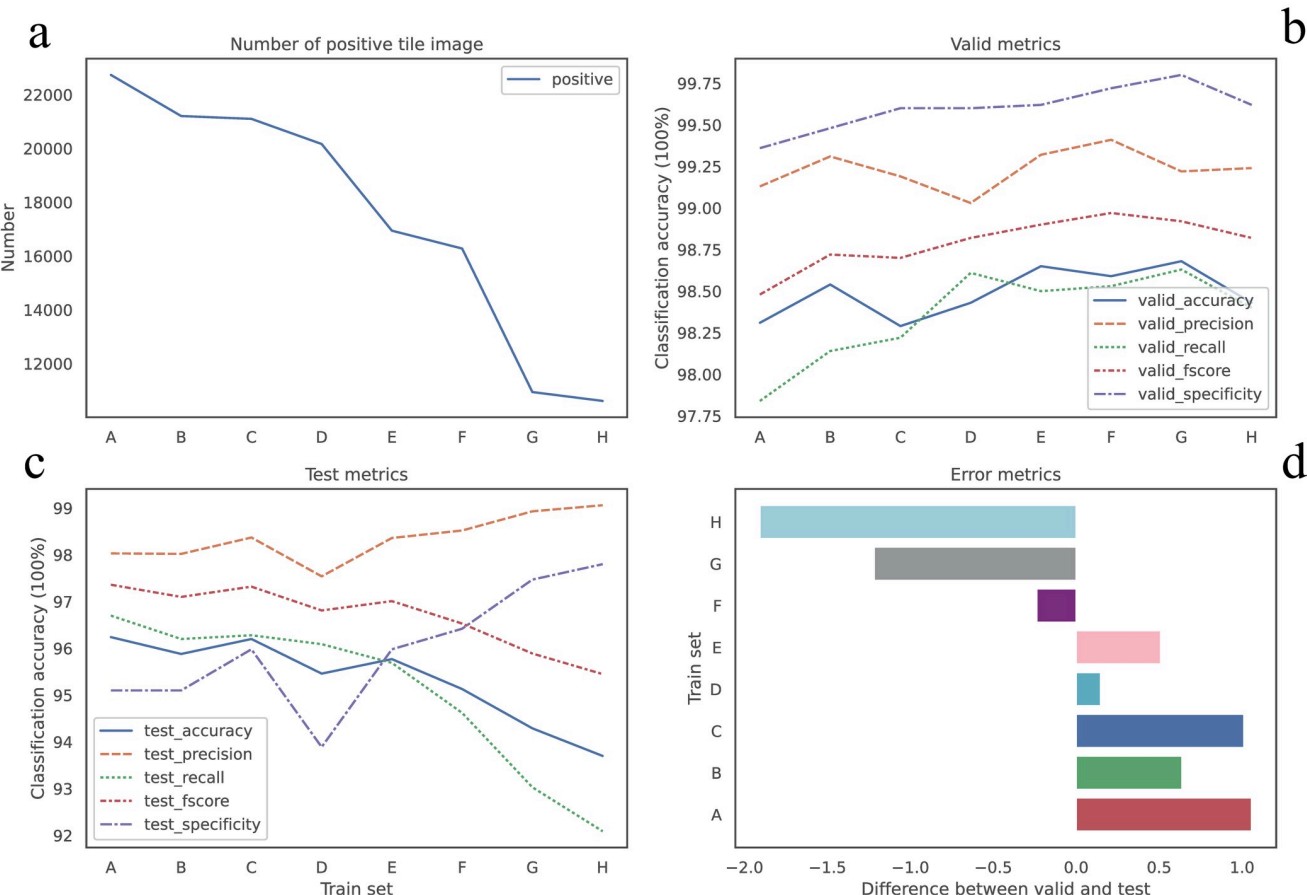

**Fig 5. Metrics for PAIP dataset.** (a) Number of the positive tile images from set A to set H is decreasing. (b)Visualize performance model on validation set. (c) Visualize performance on hold-out testing set. (d) The relative difference in the performance between validation set and test set (z-score).

Python library implementation catered to grid tiling and manually annotated files. Our study involved processing 111 digital pathology images from the CAM dataset, with an average processing time of 15.58 seconds per slide. This efficiency allows for the processing of 1000 slides in less than 5 hours, significantly accelerating the execution of downstream tasks.

In our study, we utilized the PathEX algorithm to extract tile image from two distinct data types, created eight training datasets for each type. Our experiments revealed that different *IoT* and *BoT* values have a significant impact on the downstream classification task. We found that Set C (with *IoT* and 1—*BoT* both set at 0.2) might be the optimal combination for tile image extraction from annotated files. However, we also suggest that combinations of *IoT* from 0.2 to 0.5 and 1—*BoT* from 0.2 to 0.5 could produce satisfactory results. These adjustable thresholds allow researchers to tailor the tile image extraction process to their specific studies.

In the conducted study, a comprehensive comparison of the PathEX algorithm with other extraction tools was not the primary objective. The main emphasis was placed on examining the influence of various extraction methodologies on the performance of downstream tasks and performance of the model trained on the dataset procured using the PathEX algorithm. The focal point of this investigation was predominantly on aspects related to performance, while other elements such as potential biases in models were not within the purview of this study.

As we look towards the future, there is an intention to broaden the utilization of PathEX algorithm across a more diverse array of downstream tasks. A significant objective on the horizon is development of a feature that enables the concurrent extraction of mask and tile images, a capability that could be particularly beneficial for image segmentation tasks. This enhancement would equip researchers with the capacity to employ both mask and tile images for their specific downstream tasks, thereby extending the potential applications of our algorithm. It is anticipated that these forthcoming improvements will further underscore the adaptability and efficiency of the PathEX algorithm.

## Supporting information

**S1 Table. Table of comparisons to related methods.**
(PDF)

## Acknowledgments

We appreciate the assistance from the the Artificial Intelligence Lab and the Big Data Center of Sun Yat-sen Memorial Hospital, Sun Yat-sen University.

De-identified pathology images and annotations of PAIP dataset used in this research were prepared and provided by the Seoul National University Hospital by a grant of the Korea Health Technology R&D Project through the Korea Health Industry Development Institute (KHIDI).

## Author Contributions

**Conceptualization:** Xinda Yang.

**Data curation:** Xinda Yang.

**Funding acquisition:** Kai Chen.

**Investigation:** Ranze Zhang.

**Methodology:** Xinda Yang.

**Project administration:** Yuan Yang.

**Validation:** Yu Zhang.

**Writing – original draft:** Xinda Yang.

**Writing – review & editing:** Ranze Zhang, Yu Zhang.

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
