## [Decision Letter · Decision Letter 0]

25 Mar 2024

PONE-D-24-08457PathEX: Make Good Choice for Whole Slide Image ExtractionPLOS ONE

Dear Dr. Yang,

Thank you for submitting your manuscript to PLOS ONE. After careful consideration, we feel that it has merit but does not fully meet PLOS ONE’s publication criteria as it currently stands. Therefore, we invite you to submit a revised version of the manuscript that addresses the points raised during the review process.

The reviewers have raised several comments. Please address all of them in the revised version.

We look forward to receiving your revised manuscript.

Kind regards,

Alberto Marchisio

Academic Editor

PLOS ONE

Journal Requirements:

 Whilst you may use any professional scientific editing service of your choice, PLOS has partnered with both American Journal Experts (AJE) and Editage to provide discounted services to PLOS authors. Both organizations have experience helping authors meet PLOS guidelines and can provide language editing, translation, manuscript formatting, and figure formatting to ensure your manuscript meets our submission guidelines. To take advantage of our partnership with AJE, visit the AJE website (http://aje.com/go/plos) for a 15% discount off AJE services. To take advantage of our partnership with Editage, visit the Editage website (www.editage.com) and enter referral code PLOSEDIT for a 15% discount off Editage services. If the PLOS editorial team finds any language issues in text that either AJE or Editage has edited, the service provider will re-edit the text for free.

 A clean copy of the edited manuscript (uploaded as the new *manuscript* file).

4. Please update your submission to use the PLOS LaTeX template. The template and more information on our requirements for LaTeX submissions can be found at http://journals.plos.org/plosone/s/latex.

“This work was supported by grants from the Natural Science Foundation of China (#82271650), Guangdong Science and Technology Department (2020B1212060018) and Guangzhou Science Technology and Innovation Commission (#202102010221, #20212200003).”

“the Natural Science Foundation of China (#82271650), Guangdong Science and Technology Department (2020B1212060018) and Guangzhou Science Technology and Innovation Commission (#202102010221, #20212200003).”

Reviewers' comments:

Reviewer's Responses to Questions

**Comments to the Author**

1. Is the manuscript technically sound, and do the data support the conclusions?

Reviewer #1: Yes

Reviewer #2: Yes

2. Has the statistical analysis been performed appropriately and rigorously? 

Reviewer #1: No

Reviewer #2: Yes

3. Have the authors made all data underlying the findings in their manuscript fully available?

Reviewer #1: Yes

Reviewer #2: Yes

4. Is the manuscript presented in an intelligible fashion and written in standard English?

Reviewer #1: Yes

Reviewer #2: Yes

5. Review Comments to the Author

Reviewer #1: 1.The explanation of BOT in the methodology section is lacking. Compared to IoT, obtaining BOT requires additional algorithms to calculate the area of adipocytes. The main text and pseudocode only contain textual explanations. Please provide more detailed explanations, such as adding formulas and principle explanations. Especially, if using algorithms that have already been developed, they should be referenced, and the contribution should be insufficient.

2.What is the impact of the patch extraction algorithm proposed in this study on downstream tasks? For example, common WSI classification tasks. The author should add comparative experiments in this section to demonstrate the contribution of this study

Reviewer #2: Dear Authors,

Your manuscript titled “PathEX: Make Good Choice for Whole Slide Image Extraction” presents an innovative framework for improving tile image extraction in whole slide image (WSI) analysis. The introduction of Intersection over Tile (IoT) and Background over Tile (BoT) as effective metrics to tackle specific challenges in digital pathology is noteworthy and represents a significant contribution to the field. Below are some comments and suggestions aimed at further strengthening your paper.

While the originality of PathEX are clear, and the note mentioned in the conclusion mentioning that this algorithm is not compared to other methods, the paper would greatly benefit from a more explicit comparison with existing tile extraction methods. In enhancing the comparative analysis of PathEX's performance, algorithms ranging from traditional grid tiling and state-of-the-art DL-based segmentation techniques can be utilized. Moreover, tools like Histolab and SliDL, noted for their specific functionalities in WSI analysis, also serve as good benchmarks. Additionally, evaluating PathEX alongside adaptive tiling methods and any novel, annotation-driven approaches could offer comprehensive insights into its efficacy.

The results section is well-detailed and demonstrates the efficacy of the PathEX algorithm through a variety of metrics. To further bolster these findings, consider including statistical analyses such as p-values or confidence intervals, especially when comparing the performance of different IoT and BoT settings. This would provide a stronger empirical basis for the claims made and help validate the optimal parameter configurations.

The paper touches on the implementation details and the computational resources used, which is appreciated. However, a more thorough discussion on the practical aspects of deploying PathEX, particularly its scalability to larger datasets or its integration into existing digital pathology workflows, would be valuable. Discuss any encountered limitations or challenges and how they might be overcome in future iterations of the framework.

In conclusion, your manuscript makes a valuable contribution to the field of digital pathology. Addressing these points will not only strengthen the paper but also broaden its appeal and applicability. I look forward to seeing the evolution of your work and its impact on the community.

Best regards.

6. PLOS authors have the option to publish the peer review history of their article (what does this mean?). If published, this will include your full peer review and any attached files.

Reviewer #1: No

Reviewer #2: **Yes: **Abdul Basit

---

## [Author Response · Author response to Decision Letter 0]

1 May 2024

Dear Editor and Review Committee,

I hope this letter finds you well. I am Xinda Yang, and I am writing to address the concerns raised by the reviewers in the evaluation of my research paper titled “PathEX: Make Good Choice for Whole Slide Image Extraction,” which was recently required revision.

The primary concers are the several comments from reviwers. To address these concerns, I have made some revisions to the paper. I have also provided additional information to support my claim. All the changes will be listed in the listed after the main text, which are in same order as the comments received from the reviewers.

To support my claim, I have attached supplementary information to the paper. And make some revisions to the paper to address the concerns raised by the reviewers. All the changes are marked up in the "Revised Manuscript with Track Changes" file. And the marked up color is ORANGE.

Understanding the importance of rigorous academic standards, I deeply respect the review process and appreciate the opportunity to clarify these points. I kindly request that the committee reconsider its decision, taking into account the additional information provided.

Thank you for your time and consideration. I am hopeful for a positive re-evaluation and am available for any further discussion or clarification that may be needed.

 Sincerely,

 Xinda Yang

Responses as listed below:

Respoinses to letter dated 2024-05-01

1.Please upload a Response to Reviewers letter which should include a point by point response to each of the points made by the Editor and / or Reviewers. (This should be uploaded as a 'Response to Reviewers' file type.) Please follow this link for more information: http://blogs.PLOS.org/everyone/2011/05/10/how-to-submit-your-revised-manuscript/

Response:

Point by Point response to each of the points made in the following Section "Respoinses to letter dated 2024-03-26"

1.When completing the data availability statement of the submission form, you indicated that you will make your data available on acceptance. We strongly recommend all authors decide on a data sharing plan before acceptance, as the process can be lengthy and hold up publication timelines. Please note that, though access restrictions are acceptable now, your entire data will need to be made freely accessible if your manuscript is accepted for publication. This policy applies to all data except where public deposition would breach compliance with the protocol approved by your research ethics board. If you are unable to adhere to our open data policy, please kindly revise your statement to explain your reasoning and we will seek the editor's input on an exemption. Please be assured that, once you have provided your new statement, the assessment of your exemption will not hold up the peer review process.

Response:

The data availability statement has been revised ""Respoinses to letter dated 2024-03-26". All authors agreed to make their data available upon acceptance.

1.Please ensure that you refer to Table 4 in your text as, if accepted, production will need this reference to link the reader to the Table.

Response:

The reference to Table 4 has been added to the text in manuscript.

Respoinses to letter dated 2024-03-26

https://journals.plos.org/plosone/s/file?id=wjVg/PLOSOneformattingsamplemainbody.pdf and

https://journals.plos.org/plosone/s/file?id=ba62/PLOSOneformattingsampletitleauthors_affiliations.pdf.

Response:

We checked the formatting sample and made changes accordingly, which you could find the marked up version in the "Revised Manuscript with Track Changes" file.

1.Please note that PLOS ONE has specific guidelines on code sharing for submissions in which author-generated code underpins the findings in the manuscript. In these cases, all author-generated code must be made available without restrictions upon publication of the work. Please review our guidelines at https://journals.plos.org/plosone/s/materials-and-software-sharing#loc-sharing-code and ensure that your code is shared in a way that follows best practice and facilitates reproducibility and reuse.

Response:

We have reviewed the PLOS ONE guidelines on sharing code and made sure that all author-generated code is shared in a way that follows best practice and facilitates reproducibility and reuse. All code and associated documentation are uploaded to the GitHub repository. The repository URL is https://github.com/jasnei/PathEX, which is private at the moment and added to the manuscript. We will make it publice once the manuscript is accepted. If you need reproducibility first, we could make it public at any convient. You could find the marked up version in the "Revised Manuscript with Track Changes" file. 

1.We suggest you thoroughly copyedit your manuscript for language usage, spelling, and grammar. If you do not know anyone who can help you do this, you may wish to consider employing a professional scientific editing service.

Response:

We have carefully reviewed the manuscript and made corrections to improve the language, spelling, and grammar. We have also made some minor changes to the text to improve the flow and readability. We have also added some additional explanations to the methods section to make it more clear. 

1.Please update your submission to use the PLOS LaTeX template. The template and more information on our requirements for LaTeX submissions can be found at http://journals.plos.org/plosone/s/latex.

Response:

We have updated manuscript to use the PLOS LaTeX template. 

1.Thank you for stating the following in the Acknowledgments Section of your manuscript:

"This work was supported by grants from the Natural Science Foundation of China (#82271650), Guangdong Science and Technology Department (2020B1212060018) and Guangzhou Science Technology and Innovation Commission (#202102010221, #20212200003)."

Response:

We already remove the funding information from the Acknowledgments section. We apologize for any inconvenience caused.

Comments to the Author

The following quote is the comments from Reviewer #1

Reviewer #1: 1.The explanation of BOT in the methodology section is lacking. Compared to IoT, obtaining BOT requires additional algorithms to calculate the area of adipocytes. The main text and pseudocode only contain textual explanations. Please provide more detailed explanations, such as adding formulas and principle explanations. Especially, if using algorithms that have already been developed, they should be referenced, and the contribution should be insufficient.

2.What is the impact of the patch extraction algorithm proposed in this study on downstream tasks? For example, common WSI classification tasks. The author should add comparative experiments in this section to demonstrate the contribution of this study

Response to Reviewer #1’s comments

1.Yes, obtaining BoT requires algorithms to calculate the area of adipocytes (or we call in blank area in the paper). And the BoT algorithm itself is the whole point for doing that. We already explain quiet clearly in the paper. However, there might be some not clear enough, we add how to calculate the area of the blank, which is the contour area of the mask. I hope this can dispel your doubts about BoT computation.

2.In our research, we aimed to investigate whether different combinations of IoT and BoT have an impact on downstream tasks, specifically the classification of WSI. As our experiments demonstrated, varying combinations of IoT and BoT do indeed influence WSI classification tasks, as evident in the Results section. Firstly, Higher IoT and 1 - BoT, less noise tile images, which will make the dataset is clean. More over, clean dataset (without some noisy patch images), which the classification algorithm will easy to over fit the dataset with such combination. Further more, reducing 100% blank patch images will indeed positive improve the WSI classification task. We also found out the with some noisy patch images (partially positive and partially negative patch images), which will improve the the WSI classification task model more robust in proformance.

The following quote is the comments from Reviewer #2

Reviewer #2: Dear Authors,

Your manuscript titled “PathEX: Make Good Choice for Whole Slide Image Extraction” presents an innovative framework for improving tile image extraction in whole slide image (WSI) analysis. The introduction of Intersection over Tile (IoT) and Background over Tile (BoT) as effective metrics to tackle specific challenges in digital pathology is noteworthy and represents a significant contribution to the field. Below are some comments and suggestions aimed at further strengthening your paper.

While the originality of PathEX are clear, and the note mentioned in the conclusion mentioning that this algorithm is not compared to other methods, the paper would greatly benefit from a more explicit comparison with existing tile extraction methods. In enhancing the comparative analysis of PathEX's performance, algorithms ranging from traditional grid tiling and state-of-the-art DL-based segmentation techniques can be utilized. Moreover, tools like Histolab and SliDL, noted for their specific functionalities in WSI analysis, also serve as good benchmarks. Additionally, evaluating PathEX alongside adaptive tiling methods and any novel, annotation-driven approaches could offer comprehensive insights into its efficacy.

The results section is well-detailed and demonstrates the efficacy of the PathEX algorithm through a variety of metrics. To further bolster these findings, consider including statistical analyses such as p-values or confidence intervals, especially when comparing the performance of different IoT and BoT settings. This would provide a stronger empirical basis for the claims made and help validate the optimal parameter configurations.

The paper touches on the implementation details and the computational resources used, which is appreciated. However, a more thorough discussion on the practical aspects of deploying PathEX, particularly its scalability to larger datasets or its integration into existing digital pathology workflows, would be valuable. Discuss any encountered limitations or challenges and how they might be overcome in future iterations of the framework.

In conclusion, your manuscript makes a valuable contribution to the field of digital pathology. Addressing these points will not only strengthen the paper but also broaden its appeal and applicability. I look forward to seeing the evolution of your work and its impact on the community.

Best regards.

Response to Reviewer #2’s comments

Thank you very much for your insightful comments on our research, which have highlighted some shortcomings in our paper. We have proactively adopted your suggestions and made corresponding amendments in our manuscript. Below are our responses to your comments.

Our research mainly wants to find out whether different combinations of IoT and BoT have different performances on downstream classification tasks. Therefore, we have only implemented the algorithm. Although it is an algorithm that can perform some tasks, we have not added more functions to these algorithms to become a wider-use application. And I think your suggestion is a good one, and I have added this explicit comparison to the Supporting information.

The results section do include the statistical analyses such as p-values. Such statistical analyses already included in table 3 and table 4.

Thank you for your advice. We have accepted your suggestions and diligently rewritten the discussion section, including the deployment of PathEX and its application to larger datasets. Furthermore, we have also explained the tasks that we plan to undertake in the future under the PathEX framework.

Responds to reviewer and editor comments are including in the file labeled 'Response to Reviewers'. Please do check the file. Thank you.

---

## [Decision Letter · Decision Letter 1]

17 May 2024

PathEX: Make Good Choice for Whole Slide Image Extraction

PONE-D-24-08457R1

Dear Dr. Yang,

We’re pleased to inform you that your manuscript has been judged scientifically suitable for publication and will be formally accepted for publication once it meets all outstanding technical requirements.

Please note that all the minor comments raised by the reviewers should be addressed.

Kind regards,

Alberto Marchisio

Academic Editor

PLOS ONE

Additional Editor Comments (optional):

Reviewers' comments:

Reviewer's Responses to Questions

**Comments to the Author**

1. If the authors have adequately addressed your comments raised in a previous round of review and you feel that this manuscript is now acceptable for publication, you may indicate that here to bypass the “Comments to the Author” section, enter your conflict of interest statement in the “Confidential to Editor” section, and submit your "Accept" recommendation.

Reviewer #1: All comments have been addressed

Reviewer #2: All comments have been addressed

2. Is the manuscript technically sound, and do the data support the conclusions?

Reviewer #1: Yes

Reviewer #2: Yes

3. Has the statistical analysis been performed appropriately and rigorously? 

Reviewer #1: Yes

Reviewer #2: Yes

4. Have the authors made all data underlying the findings in their manuscript fully available?

Reviewer #1: Yes

Reviewer #2: Yes

5. Is the manuscript presented in an intelligible fashion and written in standard English?

Reviewer #1: Yes

Reviewer #2: Yes

6. Review Comments to the Author

Reviewer #1: Thank you to the authors for the meticulous revisions and improvements made to the manuscript. However, there are still some minor issues that need to be addressed:

1. I recommend adding more recent references to enrich the "related work" section to ensure the timeliness and relevance of the citations. This would not only strengthen the theoretical foundation of your paper but also provide a more comprehensive overview of the latest advancements in the field. Please consider including, but not limited to, the following articles:

-Quan H, Li X, Hu D, et al. Dual-Channel Prototype Network for Few-Shot Pathology Image Classification[J]. IEEE Journal of Biomedical and Health Informatics, 2024.

-Zheng T, Chen W, Li S, et al. Learning how to detect: A deep reinforcement learning method for whole-slide melanoma histopathology images[J]. Computerized Medical Imaging and Graphics, 2023, 108: 102275.

-Ma Y, Luo X, Fu K, et al. Transformer-Based Video-Structure Multi-Instance Learning for Whole Slide Image Classification[C]//Proceedings of the AAAI Conference on Artificial Intelligence. 2024, 38(13): 14263-14271.

By referencing these publications, your paper could more fully reflect the current state of research in the field and potentially draw further inspiration for your studies.

Reviewer #2: Dear Authors,

Thank you for your resubmission of the manuscript titled “PathEX: Make Good Choice for Whole Slide Image Extraction.” I am pleased to inform you that the reviewers comments have been adequately addressed and your work represents a valuable contribution to the field.

I would like to bring to your attention a small edit required on page 9 in Table 2, where the 75th percentile heading is clipping out of the table. Please ensure this formatting issue is corrected for the final version.

Looking forward to your forthcoming publication.

Best regards.

7. PLOS authors have the option to publish the peer review history of their article (what does this mean?). If published, this will include your full peer review and any attached files.

Reviewer #1: No

Reviewer #2: **Yes: **Abdul Basit

---

## [Editor Report · Acceptance letter]

7 Jun 2024

PONE-D-24-08457R1 

PLOS ONE

Dear Dr. Yang, 

I'm pleased to inform you that your manuscript has been deemed suitable for publication in PLOS ONE. Congratulations! Your manuscript is now being handed over to our production team.

Kind regards, 

on behalf of

Dr. Alberto Marchisio 

Academic Editor

PLOS ONE